# Post-Transcriptional Regulation of Alpha One Antitrypsin by a Proteasome Inhibitor

**DOI:** 10.3390/ijms21124318

**Published:** 2020-06-17

**Authors:** Lang Rao, Yi Xu, Lucas Charles Reineke, Abhisek Bhattacharya, Alexey Tyryshkin, Jin Na Shin, N. Tony Eissa

**Affiliations:** 1Southern California Institute for Research and Education, VA Long Beach Healthcare System, Long Beach, CA 90822, USA; Tony.Eissa@va.gov; 2Department of Medicine, Baylor College of Medicine, Houston, TX 77030, USA; xuyi5510@yahoo.com (Y.X.); abhattac@alumni.bcm.edu (A.B.); alexeyt@bcm.edu (A.T.); Jinna.Shin@bcm.edu (J.N.S.); 3Department of Neuroscience, Baylor College of Medicine, Houston, TX 77030, USA; lcreinek@bcm.edu; 4School of Medicine, University of California at Irvine, Irvine, CA 92617, USA

**Keywords:** alpha one antitrypsin, alpha one antitrypsin deficiency, proteasome inhibitor, MG132, stress granule, induced pluripotent stem cells

## Abstract

Alpha one antitrypsin (α1AT), a serine proteinase inhibitor primarily produced by the liver, protects pulmonary tissue from neutrophil elastase digestion. Mutations of the *SERPINA1* gene results in a misfolded α1AT protein which aggregates inside hepatocytes causing cellular damage. Therefore, inhibition of mutant α1AT production is one practical strategy to alleviate liver damage. Here we show that proteasome inhibitors can selectively downregulate α1AT expression in human hepatocytes by suppressing the translation of α1AT. Translational suppression of α1AT is mediated by phosphorylation of eukaryotic translation initiation factor 2α and increased association of RNA binding proteins, especially stress granule protein Ras GAP SH3 binding protein (G3BP1), with α1AT mRNA. Treatment of human-induced pluripotent stem cell-derived hepatocytes with a proteasome inhibitor also results in translational inhibition of mutant α1AT in a similar manner. Together we revealed a previously undocumented role of proteasome inhibitors in the regulation of α1AT translation.

## 1. Introduction

Alpha one antitrypsin deficiency (α1ATD), a protein misfolding disorder characterized by aggregation of a misfolded alpha one antitrypsin (α1AT) protein in the hepatocytes and decreased circulating levels of α1AT, is caused by mutations of the *SERPINA1* gene [1,2]. Among the numerous variants, the Z allele (α1ATZ) causes the most common and severe forms of α1ATD. [3,4]. As α1AT protects the pulmonary tissues from the harmful effects of neutrophil elastase, patients with α1ATD often develop emphysema and chronic obstructive pulmonary disorder. Moreover, aggregation of misfolded α1AT protein in the hepatocytes may lead to cytotoxicity and cirrhosis of the liver [1]. Though an intravenous infusion of α1AT is used to treat α1ATD patients with lung diseases [1,5], this therapy does not mitigate liver damage and most patients still require liver transplantation [5,6]. Researchers have investigated several methods to reduce liver damage caused by the misfolding and aggregation of α1ATZ. These methods include gene therapy with artificial microRNA to suppress transcription of α1ATZ [7,8], induction of autophagy to remove aggregated proteins [9,10] and use of small molecules to block the polymerization of the mutant protein [11,12]. Another interesting strategy to reduce liver toxicity is to reduce the synthesis of α1ATZ by inhibiting its translation, though there has been little focus on utilizing this approach. 

The proteasome is a large protein complex that degrades damaged and misfolded proteins covalently marked with ubiquitin peptides. This intercellular hydrolysis system, also known as the ubiquitin-proteasome system (UPS), degrades more than 80% intracellular proteins and plays a major role in maintaining intracellular protein homeostasis [13,14]. As proteasome plays an essential role in regulation of important physiological processes such as the cell cycle and apoptosis, proteasome inhibitors have been extensively studied in the cancer field [15,16]. Many therapeutic strategies for treating different types of cancers utilize the ability of the proteasome inhibitors to induce cancer cell apoptosis [13,17]. PS-341 (also called Bortezomib) is one such proteasome inhibitor now approved by the FDA for treatment of multiple myeloma, and many other proteasome inhibitors are at various stages of clinical development [18,19]. Proteasome inhibition can influence protein synthesis by inducing eIF2α phosphorylation, interacting with translatome complex, and by degrading translation factors [20,21]. However, as proteasome inhibition leads to multiple downstream effects, the exact mechanism of how the proteasome regulates translation remains inconclusive. In α1ATD hepatocytes, the proteasome is partially responsible for degrading the intracellular α1ATZ aggregates [22,23]. Thus, activating the proteasome may be a useful strategy to enhance the clearance of α1ATZ whereas proteasome inhibition is likely to promote cytotoxicity. 

Here, we report that optimal concentration of proteasome inhibitors could suppress α1AT translation while minimally affecting global protein synthesis. Our study also shows that proteasome inhibitors could significantly inhibit the translation of mutant α1ATZ in induced pluripotent stem cell-derived hepatocytes.

## 2. Results

### 2.1. Alpha 1 Antitrypsin Was Down Regulated by Proteasome Inhibition

We performed our initial screening for drugs that inhibit the expression of α1ATZ in wild-type cells as both wild-type α1AT and mutant α1ATZ are transcribed and synthesized in a similar manner [1,24]. Our initial screening showed that many proteasome inhibitors substantially reduced the amount of α1AT protein and increased GADD34 in human hepatocyte cell line C3A. As stress inducible protein GADD34 is sensitive toward 26S proteasome mediated digestion [25], we thus speculate proteasome inhibition is the cause of α1AT protein reduction in those C3A cells. (Figure 1A). To rule out the possibility that the reduction of α1AT protein levels resulted from the protease inhibitory activity reported for several proteasome inhibitors, including MG32 [13], we treated hepatocytes with a protease inhibitor cocktail containing E-64 and Calpastatin. We then tried protease inhibitors mixture on hepatocytes and found no significant change on α1AT (Appendix A). However, protease inhibitor treatment did not produce any significant change in α1AT levels (Appendix A), suggesting that protease inhibitory effects of the proteasome inhibitors were unlikely to be the cause of the reduction of α1AT levels. 

To further test the role of the proteasome as a regulator of α1AT and eliminate the possible proteasome-independent side effects of the drugs, we used pre-verified siRNAs to knock down the 20S subunit of the proteasome, the target for proteasome inhibitors (Appendix A) [13]. The knockdown of the proteasome subunit 20S caused reduction of α1AT levels in C3A cells (Figure 1B). Moreover, treatment of C3A cells with lactacystin following siRNA-mediated knockdown of the 20S proteasome subunit showed further enhancement of the effects of Lactacystin in reducing α1AT levels. Though siRNA-mediated knockdown was not as efficient as the proteasome inhibitor MG132 with respect to reduction of levels of α1AT, it gave a direct evidence that proteasome could regulate α1AT production. In vivo, α1AT is mainly produced by hepatocytes and, to some extent, by pulmonary alveolar macrophages [26]. We, therefore, tested the effects of proteasome inhibitors MG132 and PS-341 on the levels of α1AT in human primary hepatocytes and human alveolar macrophages. We found that MG132 reduced steady-state level of α1AT in both human primary hepatocytes and macrophages (Appendix A). 

Finally, to confirm that proteasome inhibitors affects protein synthesis of α1AT, we used Actinomycin D to block mRNA synthesis and compared levels of α1AT in the presence and absence of MG132 in C3A hepatoma cells. Using immunoblotting and isotope labeling, we found that MG132 could inhibit α1AT synthesis similarly regardless of whether cells were treated with Actinomycin D (Figure 1C and Appendix A). Using Brefeldin A, we then blocked the secretion of α1AT and further investigated the effect of MG132 on α1AT levels. As expected, Brefeldin A completely blocked α1AT secretion by preventing its transfer from the ER to Golgi [27], and thus, only immature, non-glycosylated α1AT could be detected. However, MG132 treatment still showed a strong reduction of α1AT levels (Figure 1D). Based on these results, we conclude that proteasome inhibition reduces α1AT synthesis.

### 2.2. Proteasome Inhibitors Do Not Affect Transcription or Secretion of α1AT in Hepatocytes

To investigate whether proteasome inhibitors affect α1AT mRNA transcription, we treated C3A cells with proteasome inhibitor MG132 and then measured the mRNA levels of α1AT. We found that MG132 treatment did not alter α1AT mRNA levels as evidenced by real-time PCR (Figure 2A) or northern blotting (Appendix A). To test the effect of proteasome inhibitors on the rate of α1AT degradation, we then performed pulse-chase experiments following the treatment of C3A cells with MG132. We used Brefeldin A to block the secretion of mature α1AT from the endoplasmic reticulum and emetine to block protein synthesis. We observed less than 20% of α1AT degradation over a 4 h pulse-chase period, and there was no significant difference in the degradation of α1AT between the control and MG132 treatment was (Figure 2B). This result indicates that α1AT is a stable protein and turnover is not significantly affected by proteasome inhibition within 4 h. As a secreted protein, steady state levels of α1AT are determined by a balance between synthesis, degradation and secretion. In order to investigate the effect of MG132 on the rate of α1AT secretion, hepatocytes with or without MG132 treatment were pulsed with [^35^S] methionine/cysteine then α1AT was immunoprecipitated from cell lysates and culture media at various time points (0–30 min). Analysis of radiolabeled α1AT from cell lysate and culture media indicated mature α1AT was secreted into the media over time as expected, and no significant differences were observed between the two groups of cells in α1AT secretion (Figure 2C). These data show MG132 does not alter the rate of α1AT secretion.

### 2.3. Proteasome Inhibitors Selectively Inhibited α1AT Translation

Because proteasome inhibition did not affect α1AT transcription, degradation or secretion, we hypothesized that α1AT’s reduction caused by proteasome inhibition occurs at the translational level. To test this hypothesis, hepatocytes were pulse-labeled and α1AT levels were evaluated. Isotope labelling work indicated MG132 treatment caused significant reduction of α1AT in both mature form and immature form (Figure 3A) and this inhibition occurred even when Brefeldin A was present (Figure 3B). These results showed the inhibition of α1AT is indeed at the level of mRNA translation. Isotope labeling studies with other proteasome inhibitors PS-341 and lactacystin showed all these drugs inhibited the of synthesis α1AT, and the inhibition effect showed a time course dependent manner (Figure 3C).

To quantify the inhibition effect of proteasome inhibitors on α1AT’s translation, we pulse-labeled hepatocytes for a short period (15 min) and used Brefeldin A to block the secretion of the mature form of α1AT. Using different concentrations of MG132 we found that 10 µm of MG132 reduced α1AT synthesis by 80% (Figure 3D,E), though general protein synthesis was reduced by only 20% (Figure 3D,F). This result indicated MG132 selectively inhibited the synthesis of α1AT.

The effects of proteasome inhibitors were more pronounced on α1AT translation than that of global protein synthesis. To further examine how proteasome inhibitors affected the translation of α1AT, we treated C3A cells with MG132 (10 µm) for 4 h and performed sucrose-gradient ultracentrifugation. Then we performed qPCR to measure the levels of α1AT mRNA across the gradient. These experiments allowed us to directly examine the effects of proteasome inhibitors on the association of α1AT mRNA with the polyribosome translation complexes. Puromycin (100 µg/mL for 30 min) which dissociates polyribosome from mRNA served as a control in these experiments [28]. Our immunoblotting data showed that both puromycin and MG132 substantially reduced α1AT protein levels. However, the effects of puromycin on GADD34 and p-eIF2α levels were milder than that of MG132 (Figure 4A). In addition to decreasing the α1AT protein levels, puromycin also caused a mild decrease of polysomes and a substantial increase of free 40S, 60S ribosome subunits, and 80S ribosomes. However, MG132 treatment did not produce any gross alteration in the polyribosome pattern (Figure 4B). These results are consistent with our isotope labeling studies (Figure 3D) and further confirmed that MG132 treatment did not substantially alter global protein synthesis. We then checked the association of α1AT mRNA with different fractions of the polysome complex. Highly translated mRNAs are enriched in heavy polysomes sedimenting further into the gradients, and a shift in the mRNA to lighter fractions will be observed if translational repression occurs. We found that the in normal condition α1AT mRNA peaks are present in the heavy molecular weight fractions 15–16 (Figure 4C), indicating that α1AT mRNAs are actively translated. Both MG132 and puromycin shifted the α1AT mRNA peak to lower molecular weight polysome fraction (peak at fraction 12), signifying that both the drugs caused translational repression of α1AT mRNA (Figure 4C). The housekeeping gene β-actin did not significantly change during MG132 treatment, while puromycin treatment caused a shift to lighter molecular weight fractions (Figure 4D). These results showed that 10 µm of MG132 is sufficient to cause selective inhibition of α1AT mRNA translation without altering global mRNA translation in the cell. 

### 2.4. Phosphorylation of eIF2α Caused by Proteasome Inhibition Partially Contributing to α1AT mRNA Translation Inhibition

To investigate which signaling pathway regulated translation of α1AT mRNA during proteasome inhibition, we examined phosphorylation of eukaryotic translation initiation factor 2α (eiF2α) and eukaryotic initiation factor 4E-binding protein1 (4EBP1) as they reflect majority protein synthesis [29]. During 4 h of treatment with MG132, we observed that α1AT protein decreased constantly while phosphorylation of 4EBP1 plateaued after the first 2 h (Figure 5A). As p-4EBP1 is downstream effectors of mTOR signaling pathway [30], we thus speculated if mTOR signaling pathway has been affected by MG132 and lead to α1AT reduction. To test this hypothesis, we compared the effects of proteasome inhibitors (MG132 and PS-341) on α1AT expression with that of mTOR specific inhibitor PP242 [31]. We found that PP242 caused a significant reduction in p-S6 and p-4EBP1 but only a slight decrease of α1AT levels. In contrast, both MG132 and PS-341 caused a substantially larger reduction in α1AT, but an increase in p-S6 and p-4EBP1 (Figure 5B). Given that the increase in p-S6 and p-4EBP1 is associated with activation of mTOR and activation of cap-dependent mRNA translation [30,32], these results do not explain the effect of the proteasome inhibitors on the downregulation of α1AT. We thus concluded that the mTOR signaling pathway is not responsible for the regulation of α1AT in response to MG132. 

We also observed a gradual accumulation of GADD34 during the 4h treatment with MG132, it reflects a successful inhibition of proteasome activity (Figure 5A). Proteasome inhibitors have been reported to induce unfolded protein response (UPR) and trigger ER stress, which would induce p-eIF2α and expression of GADD34 [33]. Increasing GADD34 and p-eIF2α (Figure 5A) suggested that the ER stress response was indeed triggered in C3A cells upon proteasome inhibitors treatment. Given that the phosphorylation of eIF2α could lead to the inhibition of mRNA translation, we speculated that p-eIF2α contributed to the inhibition of α1AT expression. To evaluate this hypothesis, we overexpressed GADD34 because GADD34 could mediate dephosphorylation of eIF2α [34]. We found that overexpression of GADD34 reduced p-eIF2α and partially attenuated the reduction in α1AT following MG132 treatment (Figure 5C). In contrast, siRNA- mediated GADD34 knock down in C3A cells could promote MG132-mediated inhibition of α1AT expression and promote eIF2α phosphorylation during MG132 treatment (Figure 5D). These results suggest that there exists a correlation between the phosphorylation of eIF2α and α1AT protein synthesis.

To further test how phosphorylation of eIF2α affected α1AT protein synthesis, we used thapsigargin (a typical ER stress inducer that can induce phosphorylation of eIF2α) for a short time on C3A cells. By using immunoblotting and isotope labeling, we observed robust phosphorylation of eIF2α but only a slight inhibition of α1AT (Figure 5E,F). Furthermore, starvation and Guanabenz [35] both could trigger a strong p-eIF2α but neither altered α1AT protein levels (Appendix A). These data indicate that eIF2α phosphorylation is only partly responsible for changes in α1AT protein levels following treatment with the proteasome inhibitor.

### 2.5. Proteasome Inhibitors Increased the Association of Stress Granule Proteins with α1AT mRNA and Inhibited Translation

RNA binding proteins (RBPs) are translational regulators [36]. Proteasome inhibitors cause accumulation of RBPs in stress granules (SG) and that, in turn, inhibits protein synthesis [37]. Given that G3BP1 is the marker protein for SG evaluation [38], we treated C3A cells with MG132 and examined the formation of stress granules by detecting G3BP1 puncta formation. We found that 10 µm of MG132 could induce the assembly of SGs as early as 2 h post-treatment (Figure 6A). By using biotinylated α1AT mRNA to precipitate RBPs, we detected association of α1AT mRNA with multiple RBPs including several key SG components such as G3BP1 and G3BP2 (Appendix A). We then performed an RNA affinity pull-down assay following MG132 treatment and observed an increased association of G3BP1 with α1AT mRNA (Figure 6B). We thus hypothesized MG132 treatment could cause α1AT mRNA binding with RBPs in small SGs and inhibit mRNA translation. 

G3BP1 is a typical SGs protein, and overexpression of G3BP1 has been shown to facilitate SGs formation [39]. To confirm our hypothesis and further investigate the role for G3BP1 in regulation of α1AT mRNA translation, we overexpressed G3BP1-GFP and examine α1AT protein levels. We found that overexpression of G3BP1 by itself was not sufficient to affect α1AT levels (Figure 6C). However, compared to control cells, the effects of MG132 on the reduction of α1AT levels was more pronounced in cells overexpressing G3BP1 (Figure 6C). These data suggest that proteasome inhibition leads to an increase in the association of RNA binding proteins with α1AT mRNA and this in turn causes translational inhibition.

### 2.6. Proteasome Inhibitors Inhibited the Translation of Mutant α1ATZ in α1ATD Disease-Specific Human iPSC Derived Hepatocyte

Given that the normal α1AT and mutant α1ATZ are highly identical in primary sequence and share the same translational mechanism [24], we still want to see whether mutant α1ATZ mRNA translation could be downregulated by proteasome inhibition like the wild type counterpart. As there is very limited access to primary α1ATZ expressing liver cells, and the α1ATD patient inducible pluripotent stem cells (iPSC) derived hepatocytes has the same pathological features like the patient’s primary cells [40], we thus used human iPSC from PIZZ genotype α1ATD patients as substitute of primary α1ATZ expressing liver cells. The PIZZ iPSC and normal control cells PIMM iPSC were cultured and differentiated into hepatocyte cells which were verified by expressing hepatocyte-specific genes including albumin and alpha one antitrypsin (Figure 7A and Appendix A). We then treated the PIMM and PIZZ iPSC-derived hepatocytes with proteasome inhibitors, checked the levels of intracellular α1AT by immunoblotting, and monitored α1AT secretion by isotope labeling. Similar to our previous observations in human primary hepatocytes, we found that MG132 dramatically reduced the levels of intracellular and secreted α1AT in PIMM iPSC hepatocytes (Figure 7B). Consistent with the previous finding [40], we also detected a defect in secretion of α1ATZ into the medium in PIZZ iPSC hepatocytes. However, we did not observe any significant reduction of cellular α1ATZ levels in PIZZ iPSC hepatic cells (Figure 7B). As mutant α1ATZ could not be secreted outside of cells thus the steady state of α1ATZ detected by immunoblotting could not reflect the changes in synthesis of α1ATZ. To detect the translation of α1ATZ directly, we treated iPSC hepatocytes with proteasome inhibitors and then pulse-labeled with [^35^S] Met/Cys. Analysis of radiolabeled proteins indicated that similar to that of wild type α1AT, the translation of mutant α1ATZ in PIZZ iPSC-hepatic cells was reduced significantly (63% and 66%, respectively). Importantly overall protein synthesis decreased only around 20% (Figure 7C,D). The labeling study of iPSC hepatocytes provides a direct evidence that the proteasome inhibitors indeed selectively inhibit mutant α1ATZ translation.

## 3. Discussion

The polymerization and aggregation of mutant alpha 1 antitrypsin protein α1ATZ is the major cause of liver damage in α1ATD [4,41]. Thus, one potential solution to alleviate the underlying pathophysiology would be to block or minimize the expression of the mutant protein. Our work showed that proteasome inhibitors can substantially reduce α1AT expression. Though proteasome inhibitors can induce phosphorylation of eIF2α and reduce protein synthesis [37], we were surprised to find that proteasome inhibitors specifically inhibit translation of α1AT without grossly reducing general protein synthesis. We confirmed these specific effects of proteasome inhibitors on the translation of α1AT by using both isotopes labeling and polysome profiling assays (Figure 3 and Figure 4). These results suggest a yet unidentified role that proteasome on α1AT protein synthesis regulation.

Proteasome inhibition has been reported to influence transcription by inactivating transcription factor NF-κB [42,43]. However, this transcriptional inhibition is unlikely to contribute to the reduction of α1AT protein as the α1AT mRNA level remain unchanged upon proteasome inhibition (Figure 2 and Appendix A). Multiple lines of evidence also indicate that proteasome inhibition does not affect degradation and secretion of α1AT, at least within the 4h window we used in our study (Figure 2). Thus, we concluded that translational inhibition is the sole mechanism by which proteasome inhibition regulates the levels of α1AT protein. In eukaryotic cells two translation factors, the cap-binding complex eukaryotic initiation factor 4 (eIF4) and eIF2α regulate the majority of mRNA translation [29]. Association of eIF4 with its repressor 4EBP1 or eIF2α phosphorylation would cause inhibition of most proteins’ synthesis [29,37]. Translational reduction mediated by proteasome inhibitors has been documented in different cells [44,45] and different underlying mechanisms such as inhibiting eIF4E by inhibiting the mTOR pathway [46] and phosphorylation of eIF2α [44,47] have been proposed. Interestingly, we found that proteasome inhibitors promote phosphorylation of 4EBP1, indicating an activation of the mTOR signaling pathway. Even though the mechanism by which proteasome inhibition activates mTOR is unknown, we could conclude that the impaired synthesis of α1AT caused by proteasome inhibition is not due to mTOR inhibition.

Consistent with previous reports [37], we found that proteasome inhibition activates the unfolded protein response (UPR), and triggers phosphorylation of eIF2α and GADD34 (Figure 5A). We observed a robust increase in GADD34 shortly after the addition of proteasome inhibitors indicating the activation of UPR. We also found that p-eIF2α level peaks after 2 h of treatment and then declines slightly. This reduction is correlated with an increased in GADD34 which dephosphorylates eIF2α (Figure 5A). The dynamic changing of GADD34 and p-eIF2α levels suggest a distinct immediate feedback mechanism in hepatocytes by which these cells respond to an increase in unfolded proteins [34]. The rapid recovery from p-eIF2α could also explain why there is no significant inhibition of global protein synthesis during proteasome inhibition. 

Several studies have shown proteasome inhibition promotes eIF2α phosphorylation by influence eIF2α specific kinase like GCN2 [44], HRI [47] and PERK [48]. As PERK is an ER stress specific eiF2a kinase and proteasome inhibition induced ER stress, we predicted in our study PERK caused eiF2α phosphorylation was the major caused for α1AT suppression. However, when we apply a strong ER stress inducer TG on C3A cells treatment, it does not have the same effect as MG132 on a1AT inhibition (Figure 5E,F). All these data suggest eiF2α phosphorylation is not the sole mechanism by which proteasome inhibitors alter the translation of α1AT.

Proteasome inhibitors can inhibit mRNA translation by inducing the formation of stress granules. We thus predicted that the RBPs in the stress granules can bind to and regulate the translation of α1AT. To our satisfaction, we found that the treatment with proteasome inhibitor indeed led to stress granule formation, thereby promoting the association of multiple RBPs [39], with the mRNA of α1AT. We also found that overexpression of G3BP1 enhances proteasome inhibitor-induced α1AT translational repression. Taken together, we propose that besides causing phosphorylation of eiF2α, proteasome inhibitors could induce RBPs including G3BP1 to form stress granule complexes that selectively repress α1AT mRNA translation. 

We also showed that mutant α1ATZ from PIZZ hepatocytes was also downregulated following proteasome inhibition (Figure 7). This result indicated that proteasome which degrades soluble α1ATZ also plays an important role in regulating the translation of α1ATZ. Previous treatments of α1ATD almost exclusively relied on inducers of autophagy to remove α1ATZ aggregates, our funding of proteasome inhibitors suppress α1ATZ synthesis provide a new hint on drug development for α1ATD treatment. 

Hepatocytes are responsible for the synthesis and secretion of most plasma proteins. Downregulation of secreted proteins is one of the characteristic features of the first response toward cellular toxins [49]. In our study, we have found that MG132 treatment could significantly reduce the secretion of two other proteins, namely albumin and α fetoprotein (Appendix A). We speculate that the reduction of those proteins is also a result of translational inhibition regulated by mechanisms similar to that of α 1AT. These results suggested that any future use of the proteasome inhibitors for the treatment of α 1ATD might result in a reduction of the concentration of hepatic secretory proteins such as albumin and α-fetoprotein in the blood circulation. Given that some proteasome inhibitors have already been widely used in cancer treatment [18], this finding suggests that there might be a reduction of the concentrations of hepatic secretory proteins in the circulation as these proteasome inhibitors could inhibit the synthesis of these proteins. 

## 4. Materials and Methods

### 4.1. Reagents Cells and Antibodys

MG132, PS-341, Lactacystin, Calpastatin, Brefeldin A, emetine, puromycin, penicillin, streptomycin, Guanabana and Thapsigargin were purchased from Sigma (St. Louis, MO, USA). 

Normal human hepatocyte and human Macrophage were obtained from Lonza, Hepatocellular cell line C3A and ubiquitin-proteasome system reporter cell line GFPu-1 were from ATCC, and these hepatocytes were cultured in DMEM/F12 1:1 medium supplemented with 10% fetal bovine serum and 1% penicillin-streptomycin. The α1ATD disease specific Human induced pluripotent stem cells were provided by David A. Lomas. The mutant and wide type iPSC cells lines were differentiated in definitive endoderm over 5 days (T5), and then into early hepatocytes over 24 days following standard protocol. The successfully differentiated hepatocytes were then confirmed by α1AT’s gene expression using qPCR. Mature hiPSC hepatocytes cultured in Hepatocyte Culture Medium (GIBCO, Columbia, USA).

Polyclonal Rabbit anti-Human alpha-1-antitrypsin (A0012) was purchased from Biocompare. Antibodies for G3BP1 and GADD34 were purchased from Proteintech and antibodies for eIF2α, p-eIF2α, p-S6, S6, p-4EBP1 and 4EBP1 were purchased from Cell Signaling Technology (Danvers, MA USA).

### 4.2. qPCR, Northern Blot Analysis and Western Blot Analysis

Total RNAs were extracted from cells with RNA purification kit (Qiagen, Hilden, Germany) according to the manufacturer’s protocol. For qPCR analysis, mRNA was first reverse transcribed into cDNA using High Capacity cDNA archiving kit (Applied Biosystems, Foster City, CA, USA). Relative expressions of specific genes were calculated by ΔΔCT method by normalizing to the expression of glyceraldehyde phosphate dehydrogenase (GAPDH). Primers for the assay: ALB: F 5′-cacaatgaagtgggtaacc-3′, R 5′-gacatcctcagcttgctgtc-3′ AFP: F 5′-agaacctgtcacaagctgtg-3′, R 5′-tggtagccaggtcagctaaa-3′; GAPDH: F 5′-gacttcaacagcgacacc-3′, R 5′-tagccaaattcgttgtcata-3′; HNF-4α: F 5′-gggtgggcttggccatgg-3′, R 5′-gggccgctgggcagagtt-3′,α1AT: F 5′-ggctgacactcacgatgaaa-3′, R 5′-gtgtccccgaagttgacagt-3′. For Northern blot assay: Total cellular RNAs were separated by electrophoresis on 1.5% agarose formaldehyde gels and transferred to nitrocellulose membranes. The membranes were hybridized with ^32^P-labeled cDNA probes and exposed to autoradiographic film. 

Cells with or without drugs treatment were lysed with appropriate amount of lysis buffer (40 mm Bis-Tris propane, 150 mm NaCl, 10% glycerol and 1% Triton X-100), supplemented with Proteinase Inhibitors Cocktail (Roche Applied Sciences, Penzberg, Germany) and Halt Phosphatase Inhibitor Cocktail (Thermo Fisher Scientific, Waltham, MA, USA) at 4 °C. Cellular pellets were removed by high speed centrifugation 12,000 × rpm for 10 min. Protein concentration for each sample were evaluated with Pierce BCA protein assay kit. Equal amounts of proteins (50 µg) were applied on SDS-PAGE. Separated proteins were transferred to nitrocellulose membrane using semidry transfer cell (Bio-Rad, Hercules, CA, USA). Membranes were blotting with primary antibody and subsequently secondary antibodies. Immunoreactive protein bands were detected by Odyssey Imaging System (LI-COR Bioscience, Lincoln, NE, USA).

### 4.3. Polysome Profiling by Sucrose Gradient Fractionation 

Hepatocytes were seeded with 3 × 10^6^ in each 100 mm dish, cultured for overnight and then applied with or without drugs treatment. Before cell harvesting, cycloheximide were added to 100 µg/mL treated for 10 min and washed with ice-cold PBS containing cycloheximide buffer twice. Polysome lysates were produced by on plate lysis with appropriate volume of Polysome extraction buffer (20 mm Tris-HCl pH 7.5, 100 mm KCl, 5 mm MgCl_2_, 1% Triton X-100, 0.1 mg/mL cycloheximide, 2 mm DTT, 100~200 units/mL RNAse inhibitor, EDTA-free protease inhibitors). With a centrifugation of 12,000 × rpm for 10 min at 4 °C, the clear supernatant was layered on 10–50% continuous sucrose gradient. The samples were then ultracentrifugated at 35,000 rpm in a SW41-Ti rotor (Beckman, Indianapolis, IN, USA) at 4 °C for 2 h. The samples were separated into 17 fractions by ISCO Foxy Jr. gradient fractionator (Teledyneisco, Lincoln, NE, USA) with continuous measurement of the absorbance at 254 nm by UA-6 Detector. After collecting fractions, spike-in control luciferase mRNA L4561 (Promega, Madison, WI, USA) was add into each sample to a final 10 ng/mL. RNA from each fraction of polysome analysis was isolated using Trizol reagent (Invitrogen, Carlsbad, CA, USA) and cDNA was synthesis using High Capacity cDNA archiving kit (Applied Biosystems, Foster City, CA, USA). Applied Biosystems Real-Time PCR Systems was used to quantify transcript abundance for genes of interest with luciferase (Promega, Madison, WI, USA) as internal control. The primers used for qPCR are as follow α1AT: F 5′-ggctgacactcacgatgaaa-3′, R 5′-gtgtccccgaagttgacagt-3′; β-actin: F 5′-tgacaggatcgagaaggaga-3′, R 5′-cgctcaggaggagcaatg-3′; luciferase: F 5′-ccagggatttcagtcgatgt-3′, R 5′-aatctcacgcaggcagttct-3′. 

### 4.4. RNA Interference-Mediated Gene Knockdown and Plasmids Transfection

Pre-validated short interfering siRNAs Proteasome 20s subunit PSMB2 s481 (Thermo Fisher Scientific, Waltham, MA, USA), GADD34 PPP1R15A s24268 were obtained from Ambion. For RNA transfection, hepatocyte cells were first seeded in 6-well plates at a density of 5 × 10^5^ cells /well. The next day, in each well the cells were transfected with 20 nm siRNA pool using 10 uL of Lipofectamine 2000 reagent (Invitrogen, Carlsbad, CA, USA) according to the manufacturer’s instructions. Forty-eight hours post transfection, cells were either treated with drugs for indicated times or harvested for immunoblot analysis. Plasmids transfection followed the same method as siRNA transfection using 10 μL of lipofectamine 2000 and 2 μg of plasmid DNA.

### 4.5. Radiolabeling Study

In total, 1 × 10^6^ cells were seeded on 60 mm tissue culture dishes with complete medium DMEM and incubated overnight. The next day each plate of cell was treated with indicated drugs for 1 h, then change the medium with methionine/cysteine free DMEM for 30 min then the Express [^35^S] Protein Labeling Mix (Perkin Elmer, Waltham, MA, USA) (100 µCi/mL; 1000 Ci/mmol) was added to pulse for 15 min. After washing with PBS, each plate was added with lysis buffer for cell harvest. To evaluate the gross protein synthesis: the same amount of total cell lysates was loaded to SDS-PAGE for gel electrophoresis. Then the gel was exposed to a phosphor-imager plate to detect the radiation intensity. To evaluate specific protein’s synthesis rate, 2 μg antibody of each protein was added into isotope labelled cell lysates with total of 1mg protein. The cells lysate with antibody were transferred into a 1.5 mL tube and rotated for overnight 4 °C, and then added with 20 μL of Protein A/G Magnetic Beads (Thermo Fisher Scientific, Waltham, MA, USA) and rotated for another 2 h. The tube was then placed into a magnetic stand to collect the beads. The beads were washed 3 times using 400 μL lysis buffer and then resuspended into 30 μL elution buffer. The elution with target protein was adjusted to neutral pH with neutralization buffer and then applied on SDS-PAGE for protein separation. Finally, the gel was exposed to phosphor-imager to detect the radiation intensity. 

To determine α1AT’s degradation process, the cells were first radiolabeled for 1 h, and then changing the medium with unlabeled media containing emetine 10 μM and Brefeldin A 2 µg/mL. The cells then were chased for different time point and in each sample the target proteins were collected, and radio signal was detected using the same method as described upon.

### 4.6. Plasmids Construction

The total cDNA was synthesized using SuperScript^®^ III First-Strand Synthesis System (Thermo Fisher, Waltham, MA, USA) with total mRNA extracted from C3A cells as template. The α1AT coding sequence with was amplified by PCR using primers with HindIII and XhoI restriction sites: α1AT F 5′-aagcttactcagtaaatggtagatc-3′ and α1AT R 5′-gtaaatggtagatc-3′. The PCR product was purified and digested with HindIII and XhoI restriction endonucleases (NEB) and inserted into pcDNA3.1 (Invitrogen, Carlsbad, CA, USA) cut by the same endonucleases to produce a recombinant plasmid pcDNA3.1α1AT. Plasmid express GFP-tagged human GADD34 [pEGFP-C1GADD34] obtained from Dr.Shirish Shenolikar’s lab (Brown University, Providence, RI, USA). Plasmids G3BP-GFP-λN and GFP-λN obtained from Dr Reineke LC (Baylor college of Medicine, Houston, TX, USA).

### 4.7. Biotinylated RNA Pull-Down Assay

To prepare the biotin-labeled RNA, 2 ug plasmid pcDNA3.1 α1AT were first linearized with NheI, purified and then transcripted by T7 RNA polymerase (NEB, Ipswich, MA, USA) with Biotin-RNA labelling Mix (Roche, Basel, Switzerland). The Biotin labelled RNA was then digested with RNase-free DNase and purified with RNA Cleanup Kit (Qiagen, Hilden, Germany). For biothylated RNA pull down, C3A cells were first lysed by extraction buffer (20 mm Tris-HCl, pH 8.0, 150 mm NaCl, 5% glycerol, 0.1% Triton X-100, 1 mm DTT, 1 mm PMSF and protease Inhibitor) the nuclear and debris were pelleted by centrifugation at 13,000 rpm for 20 min. In each assay 2 μg of biotinylated transcript were introduced into 500 ug of C3A cell extract incubated and shaking in 4 °C for 1 h; 20 μL of washed Streptavidin agarose beads (Invitrogen, Carlsbad, CA, USA) were added to each binding reaction and further incubated at RT for 1 h. Beads were then washed briefly three times with lysis buffer and collected by centrifugation, and then were loaded on SDS-PAGE gels, and the corresponding band was sent for mass spectrometry analysis.

### 4.8. Immunofluorescence Staining

Cells plated on type I collagen-coated glass coverslips were washed twice with PBS, fixed with 4% paraformaldehyde and permeabilized with 0.2% Triton X-100 10 min. After incubated first with G3BP1 antibody for 1 h and then with goat anti-rat IgG conjugated to Alexa Fluor 488, the coverslips were mounted on microscope slides with ProLong Gold Antifade reagent. Images were acquired using a Zeiss Axiovert 200 M microscope (Artisan, Champaign, IL, USA).

## Figures and Tables

**Figure 1 ijms-21-04318-f001:**
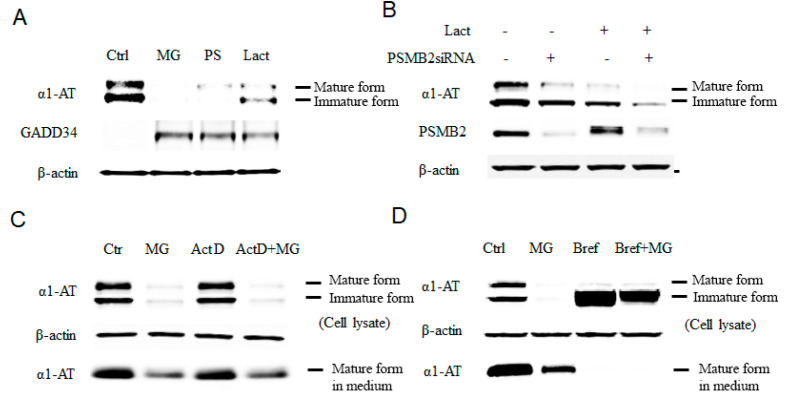
Inhibition of proteasome causes reduction of the alpha 1 antitrypsin (α1AT) level in human hepatocyte cell line. (**A**) C3A cells were incubated for 4 h in the presence or absence (control) of proteasome inhibitors MG132 (10 µm), PS-341 (10 µm), or lactacystin (10 µm). (**B**) Knock down of the 20 s proteasome subunit β2 was performed in with siRNA for 72 h and cells were then treated with or without lactacystin (10 µm) for 4 h. (**C**) C3A cell were treated for 4 h with actinomycin D (0.1 μg/mL), MG132 (10 µm) or both actinomycin and MG132. (**D**) C3A cells were incubated for 4 h in the presence or absence of MG132 (10 µm), Brefeldin A (2 µg/mL) or both Brefeldin A and MG132. Cell lysates were analyzed by western blot analysis using antibodies to α1AT, GADD34 or β-actin.

**Figure 2 ijms-21-04318-f002:**
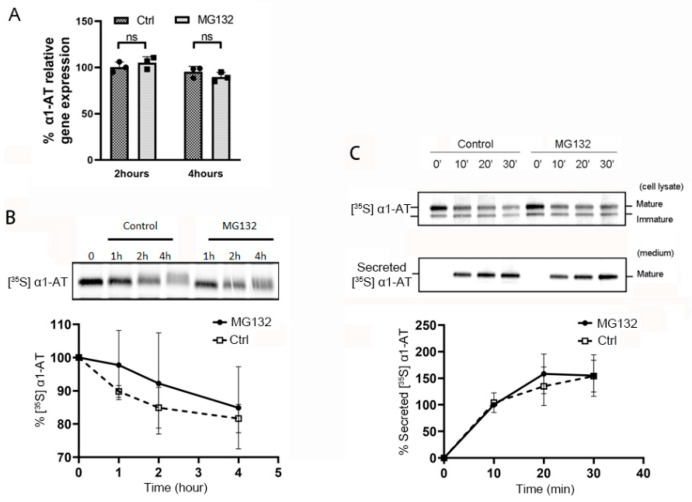
Proteasome inhibitor MG132 does not alter α1AT mRNA level, degradation of α1AT protein or its secretion during 4 h of treatment. (**A**) C3A cells were incubated for 2 or 4 h in the presence or absence of MG132 (10 µm). Levels of α1AT mRNA was evaluated by qPCR. (**B**) C3A cells were pulsed with [^35^S] methionine/cysteine for 1 h and chased with unlabeled media at various time points in the absence or presence of MG132 (10 µm). During the chase, translation and secretion of α1AT were blocked by emetine 10 µm and Brefeldin A (2 µg/mL) respectively. α1AT was immunoprecipitated and eluted proteins were analyzed by SDS/PAGE. Bands representing [^35^S] labeled α1AT were quantitated using phosphor-imager. Data are means ± SD, *n* = 3. (**C**) Hepatocyte C3A cells were first pulsed with [^35^S] methionine/cysteine for 1 h and chased with unlabeled medium at various time points in the absence or presence of MG132 (10 µm). α1AT was immunoprecipitated from both cell lysates (upper panel) and culture media (lower panel) were analyzed by autoradiography. Bands representing [^35^S] labeled α1AT in culture media (lower panel) were quantitated by phosphor-imager. Data are means ± SD, *n* = 3.

**Figure 3 ijms-21-04318-f003:**
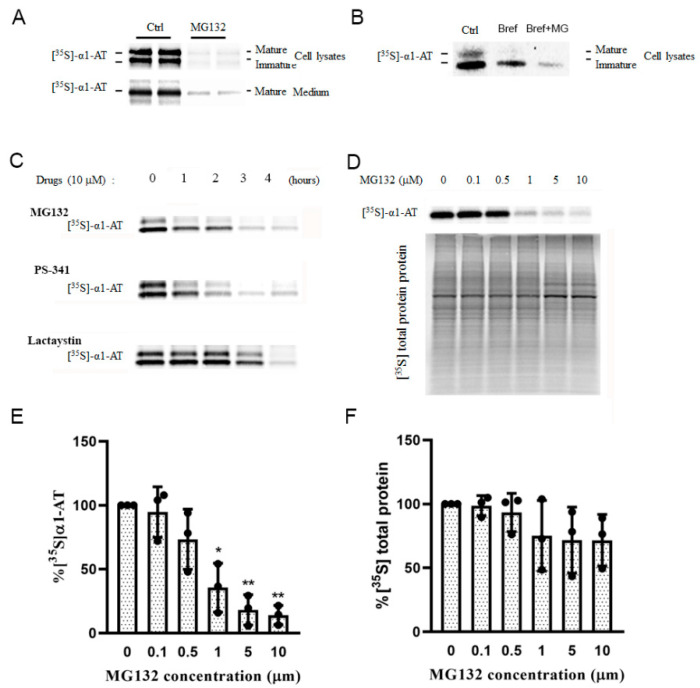
Proteasome inhibitors inhibited α1AT synthesis in hepatocytes. (**A**) C3A cells were subjected to none or 10 µm MG132 treatment for 4 h and then labeled with [^35^S] methionine/cysteine for 30 min; (**B**) Brefeldin A (2 µg/mL) was added to the medium. α1AT was immunoprecipitated from cell lysates and [^35^S] labeled α1AT was detected by autoradiography. (**C**) C3A cells were incubated with 10 µm of MG132, PS-341 and Lactacystin for different time points (0 to 4 h) and the cells were labeled with [^35^S] methionine/cysteine for additional 30 min. α1AT was immunoprecipitated from cells lysates and culture media. Eluted proteins were analyzed by SDS-PAGE and [^35^S] labeled α1AT was detected by autoradiography. (**D**) C3A cells were incubated with different concentration of MG132 for 4 h. Cell were then labeled with [^35^S] methionine/cysteine for 15 min. [^35^S] labeled α1AT and general protein were detected by autoradiography. Isotope labelled α1AT (**E**) and total protein (**F**) were quantitatively analyzed; data represent means ± SD of three independent experiments. * *p* < 0.05; ** *p* < 0.001.

**Figure 4 ijms-21-04318-f004:**
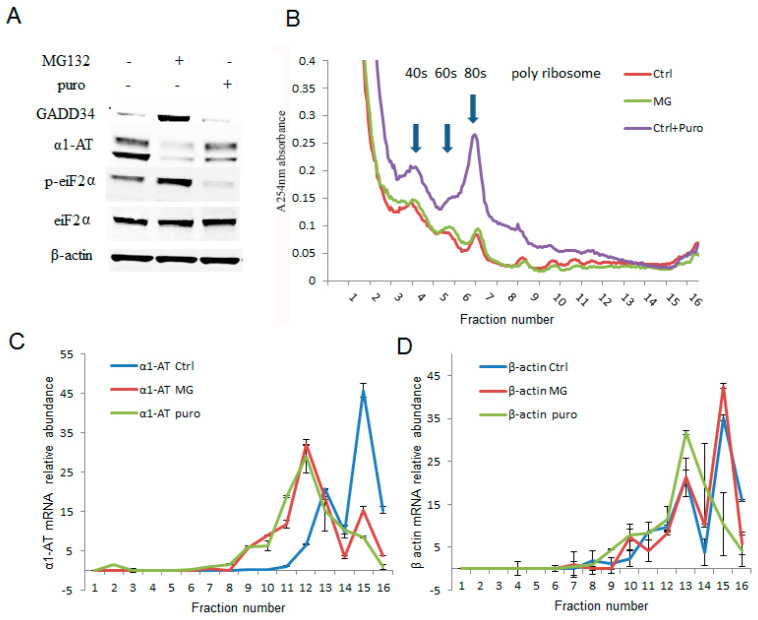
Proteasome inhibitor MG132 inhibited α1AT mRNA translation. (**A**) C3A cells were treated with MG132 (10 µm) for 4 h and puromycin (2 µg/mL) for 30 min. The steady state level of GADD34, α1AT, p-eIF2α, eIF2α and β-actin were detected by western blotting using specific antibodies. (**B**) C3A cells lysates from (**A**) were fractionated on 10–50% sucrose gradients, 40S, 60S and 80S ribosomal subunits (fraction 3–7) as well as polysomes (fraction 8–16) were measured by detecting absorbance at 254 nm. (**C**,**D**) mRNA of α1AT (**C**) and β-actin (**D**) in each gradient fraction were measured by quantitative real-time PCR. mRNA level was calculated by spiked internal control and plotted as a percentage of the total mRNA levels in that sample. Data represent the mean of three independent experiments ± SD.

**Figure 5 ijms-21-04318-f005:**
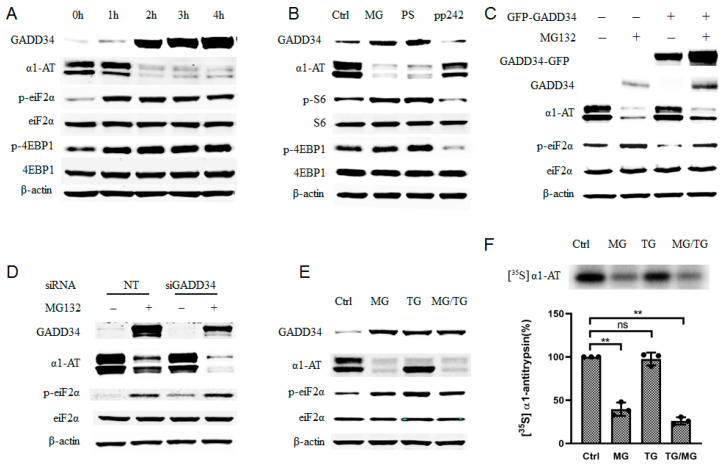
Proteasome inhibitor induced phosphorylation of eIF2α only partially contributed to α1AT translation inhibition. (**A**) C3A cells were treated with MG132 (10 µm) for varying duration (0–4), time-dependent effect of MG132 on phosphorylation of eIF2α and 4EBP1 was detected. (**B**) C3A cells were treated with proteasome inhibitors MG132 (10 µm), PS-341 (10 µm) and mTOR inhibitors PP242 (0.5 µm) for 4 h. (**C**) C3A cells were transiently transfected with empty vector or GFP tagged GADD34 for 48 h and then treated with MG132 for 4. (**D**) C3A cells were transfected with GADD34 siRNA (100 µm) for 48 h, and then treated with or without MG132 for 4 h. (**E**) C3A cells were treated with MG132 (10 µm) for 4 h and thapsigargin (5 µm) for 2 h, cells lysates applied for western blot with indicated antibody. (**F**) C3A cells were treated with MG132 for 4 h, or with thapsigargin for 2 h and then labelled with [^35^S] methionine/cysteine for 30 min. Cells lysate was then applied with immunoprecipitation using α1AT antibody. [^35^S] labeled α1AT was detected by autoradiography, data represent means ± SD of three independent experiments. ** *p* < 0.001.

**Figure 6 ijms-21-04318-f006:**
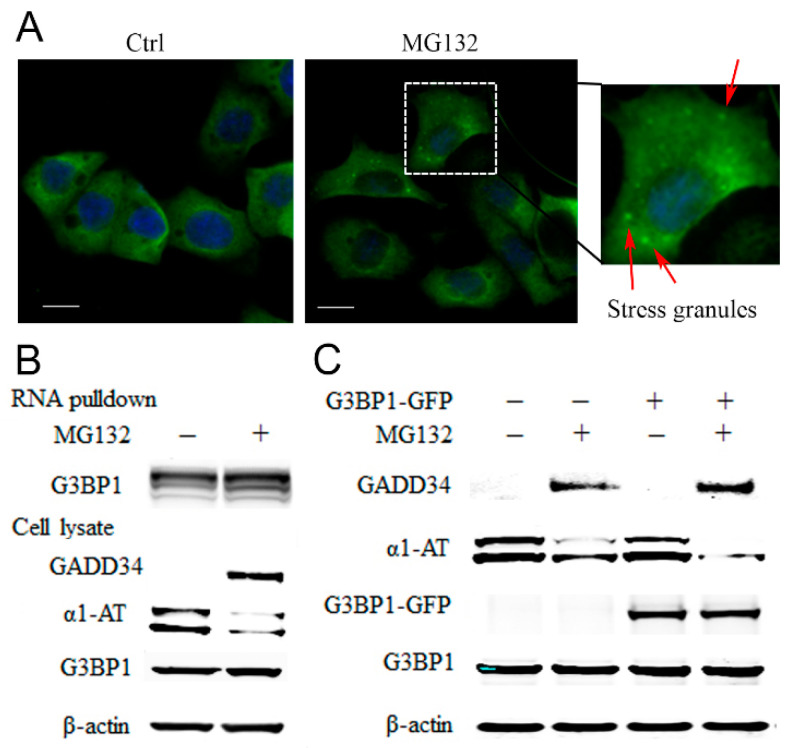
MG132 causes stress granule formation and promotes RNA binding proteins associated with α1AT mRNA. (**A**) C3A cells were plated on type I collagen-coated glass coverslips and treated with MG132 for 2 h. The cells were stained with G3BP1 antibody and mounted using the Prolong Gold Antifade reagent with DAPI. The positive G3BP1 puncta which represent stress granule were denoted by an arrow (Scale bar = 10 μm). (**B**) Biotinylated α1AT RNA incubated in 500 µg C3A cytoplasmic protein (Ctrl and MG132 treatment). The RNA-RBPs complex were pulled down with streptavidin agarose beads and applied for western blot analysis. (**C**) C3A cells were transiently transfected with empty vector or GFP tagged G3BP1 for 48 h and then treated with MG132. Cell lysates were analyzed by western blot analysis using indicated antibodies.

**Figure 7 ijms-21-04318-f007:**
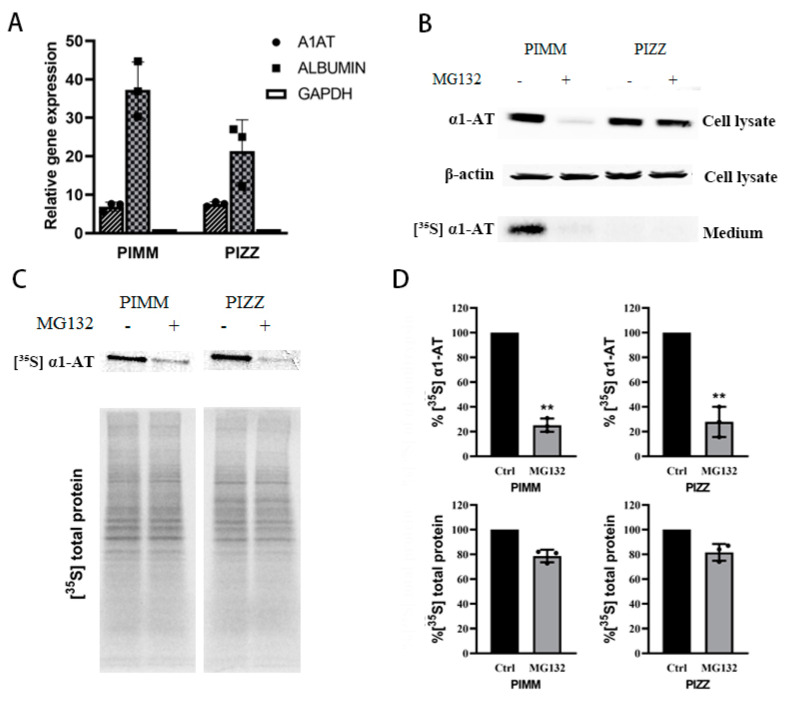
Proteasome inhibitors reduced wild type and mutant α1AT expression in human iPSC-derived hepatocytes. (**A**) Normal (PIMM) and α1ATD patient (PIZZ) inducible pluripotent stem cells were differentiated into hepatocytes; cellular albumin and α1AT mRNA were evaluated by qPCR. (**B**) PIMM and PIZZ iPSC-hepatic cells were treated with or without MG132 (10 µm) for 4 h and labeled with [^35^S] methionine/cysteine for 30 min. The steady stage of synthesized α1AT (PIMM) and α1ATZ (PIZZ) were detected by western blotting. Secreted α1AT from culture media was immunoprecipitated and detected by autoradiography. (**C**, **D**) [^35^S] labeled synthesized cellular α1AT, α1ATZ and general protein were detected by autoradiography and quantitatively analyzed. Data represent means ± SD of three independent experiments. ** *p* < 0.001.

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
