# Peer review of "Post-Transcriptional Regulation of Alpha One Antitrypsin by a Proteasome Inhibitor"

_ijms, 2020, doi:10.3390/ijms21124318_

Round 1
Reviewer 1 Report
The paper by Rao, et al describes the discovery and characterization of translational control of the alpha one antitrypsin protein (1AT). The protein is synthesized in the liver and secreted to circulate to the lungs were it plays an important protective role by inhibiting the elastin protease. Synthesis and secretion of the protein is impaired by cancer therapies that utilize the 26S ubiquitin proteasome inhibitor MG132 (MG). The authors describe how 1AT mRNA translational control correlates with the unfolded protein response (UPR) and eIF2 phosphoryation, which are both induced by MG.
Unfortunately, the authors fail to address the most important regulator at the center of UPR-mediated eIF2 regulation, the PERK kinase. This is a glaring omission in this paper, as PERK resides in the ER where the UPR response occurs AND where 1AT must occur on rough ER ribosomes to allow it to be secreted. They further overlook the differences between protein synthesis regulation for soluble ribosomes and those on the RER that mediate secreted protein synthesis. The manuscript is fraught with grammatical errors and lacking in reference to the well-studied UPR regulation of secreted protein synthesis.
Specific Comments:
1. The entire experimental study ignores the PERK kinase, which has long been known to mediate UPR inhibition of secreted protein synthesis by directly phosphorylating eIF2 (Harding, et al, 2000; Walter and Ron, 2011). This indicates a poor understanding of the field of secreted protein synthesis by the authors.
2. The study overlooks the known differences between protein synthesis regulation for soluble ribosomes and those on the RER that mediate secreted protein synthesis. The fact that mRNAs segregate/compete quite differently for translation on soluble ribosomes and those that associate with ER has been known for nearly four decades (Richter and Smith, 1981). Therefore, when the authors assay actin synthesis (and actin mRNA on polysomes, Fig. 4) made on soluble ribosomes as a control for 1AT protein synthesis (made on RER), they are comparing apples and oranges.
3. The authors naively claim that 1AT regulation by MG is an mRNA-specific response to UPR by comparing its synthesis to actin throughout. In fact, it is likely that the same regulation by the MG proteasome inhibitor is occurring for all mRNAs encoding secreted proteins in their cells. In fact, since PERK kinase resides with the ER membrane and directly binds unfolded proteins to become activated, its effects are likely to be most pronounced for secreted protein synthesis. However, because the authors never assay another secreted protein in any of the presented experiments as a control, they have greatly overinterpreted the single finding.
4. The PERK kinase has another direct link to the proteasome in that its activity enhances the degradation of known unstable proteins like cyclins by upregulating proteasome activity directly (Raven and Koromilas, 2008).
5. There are countless grammatical and vocabulary errors in the entire manuscript. For example, in lines 11-15, seven grammatical errors were found. The manuscript should be edited by someone with language training or a native English speaker prior to submission.
6. The Methods are poorly described and contain insufficient information on a procedure. Figure 4 utilizes sucrose gradient resolution of polysomes. The text says a 15-50% gradient was used, but the Methods say a 10-50% gradient was used. It says very little else. It is important to know what size gradient/SW rotor was used, how long it was centrifuged and at what g-force. The conditions might indicate that the author’s over-sedimented their gradients, and have pushed most polysomes over 7-mers to a pellet at the bottom of the tube. This may explain why all the mRNAs tested are low in fraction 14 and spike high in fraction 15.
7. Treatment of cells prior to the polysome analysis is also problematic. The authors incubate cells 10 min in cycloheximide prior to lysis. This pushes even poorly translating mRNAs into dense, arrested polysomes and gives anomalously “heavy” polysomes, which is why their 1AT mRNA appears in 6-7-mer polysomes even when eIF2 is phosphorylated after MG treatment. It may also explain why puromycin, which should dissociate all polysomes when fed to cells, has just a modest effect on polysome size. The authors should also explain why both MG and puromycin cause the same reduction of 1AT polysome size (Fig. 4C) but MG lowers 1AT protein so much more than puromycin does (Fig 4A).
8. Some Methods are completely absent. For instance, the method used to compare 1AT mRNA levels by qRT-PCR in Fig 2A are completely absent. What mRNA was used as a standard to normalize? Did the authors use a 2EdeltadeltaCt formula, as is conventional?
References:
1. Harding, H. P., Novoa, I., Zhang, Y., Zeng, H., Wek, R., Schapira, M. and Ron, D. (2000). Regulated translation initiation controls stress-induced gene expression in mammalian cells. Mol Cell 6, 1099-108.
2. Walter, P. and Ron, D. (2011). The unfolded protein response: from stress pathway to homeostatic regulation. Science 334, 1081-1086.
3. Raven, J. F. and Koromilas, A. E. (2008). PERK and PKR: old kinases learn new tricks. Cell Cycle 7, 1146-1150.
4. Richter, J. D. and Smith, L. D. (1981). Differential capacity for translation and lack of competition between mRNAs that segregate to free and membrane-bound polysomes. Cell 27, 183-191.
Author Response
The paper by Rao, et al describes the discovery and characterization of translational control of the alpha one antitrypsin protein (1AT). The protein is synthesized in the liver and secreted to circulate to the lungs were it plays an important protective role by inhibiting the elastin protease. Synthesis and secretion of the protein is impaired by cancer therapies that utilize the 26S ubiquitin proteasome inhibitor MG132 (MG). The authors describe how 1AT mRNA translational control correlates with the unfolded protein response (UPR) and eIF2 phosphoryation, which are both induced by MG.
Unfortunately, the authors fail to address the most important regulator at the center of UPR-mediated eIF2 regulation, the PERK kinase. This is a glaring omission in this paper, as PERK resides in the ER where the UPR response occurs AND where 1AT must occur on rough ER ribosomes to allow it to be secreted. They further overlook the differences between protein synthesis regulation for soluble ribosomes and those on the RER that mediate secreted protein synthesis. The manuscript is fraught with grammatical errors and lacking in reference to the well-studied UPR regulation of secreted protein synthesis.
Specific Comments:
- The entire experimental study ignores the PERK kinase, which has long been known to mediate UPR inhibition of secreted protein synthesis by directly phosphorylating eIF2 (Harding, et al, 2000; Walter and Ron, 2011). This indicates a poor understanding of the field of secreted protein synthesis by the authors.
Response: We appreciate reviewer’s comment. However, our central thesis is that 1AT mRNA translation is regulated differently under conditions of proteasome inhibition and eIF2a phosphorylation. We acknowledge that PERK is likely activated under our conditions, but other kinases have also been shown to be activated under proteasome inhibition such as GCN2 (Jiang & Wek, 2005) and HRI (Yerlikaya et al, 2008). In our study, the status of eIF2α phosphorylation reflects the sum of activation of all possible kinases (including PERK).
Indeed, the effect on 1AT mRNA translation differs when we directly stress the ER with thapsigargin further illustrating the sensitivity and specificity of 1AT mRNA regulation to proteasome inhibition (Figure 5F). It seems that MG132-dependent regulation of 1AT synthesis is not solely depend on ER stress induction of eiF2α phosphorylation. Our study shows proteasome inhibitors cause RNA binding protein association with a1AT mRNA and contributes to a1AT’s translation suppression (Figure 6). Nevertheless, we have added more discussion about the potential kinases in revised manuscript.
- The study overlooks the known differences between protein synthesis regulation for soluble ribosomes and those on the RER that mediate secreted protein synthesis. The fact that mRNAs segregate/compete quite differently for translation on soluble ribosomes and those that associate with ER has been known for nearly four decades (Richter and Smith, 1981). Therefore, when the authors assay actin synthesis (and actin mRNA on polysomes, Fig. 4) made on soluble ribosomes as a control for 1AT protein synthesis (made on RER), they are comparing apples and oranges.
Response:We appreciated reviewer for bringing up this point.
Both soluble and ER-bound ribosomes are subject to regulation by eIF2a phosphorylation. As our studies using s35 methionine labelling has shown α 1AT is repressed much more strongly than general protein synthesis, here we just want to use the polysome profiling data to confirm the previous finding.
It might be true a1AT’s dramatic translation inhibition is a result of ER bonded ribosome protein synthesis suppression caused by ER stress. But it does not explain why thapsigargin which is a typical ER stress inducer causing stronger eIF2a phosphorylation but does not have the same effect as MG132 (Figure 5F). All of those data suggested MG132 caused a1AT suppression might be more complicated than ER stress response.
- The authors naively claim that 1AT regulation by MG is an mRNA-specific response to UPR by comparing its synthesis to actin throughout. In fact, it is likely that the same regulation by the MG proteasome inhibitor is occurring for all mRNAs encoding secreted proteins in their cells. In fact, since PERK kinase resides with the ER membrane and directly binds unfolded proteins to become activated, its effects are likely to be most pronounced for secreted protein synthesis. However, because the authors never assay another secreted protein in any of the presented experiments as a control, they have greatly overinterpreted the single finding.
Response: As mentioned above, both α1AT and actin mRNAs are subject to repression under conditions of eIF2a phosphorylation. The isotope labelling work clearly shows that 1AT is more sensitive than the majority of proteins being synthesized within the cell. We DID NOT claim protein synthesis of 1AT is single gene specific. In fact, we agree that other mRNAs may be affected. We have also found other secreted proteins like album and α-fetoprotein are down regulated by MG132 treatment (figure bellow), as our project is focus on the α1AD treatment, we didn’t include that in our discussion.
- The PERK kinase has another direct link to the proteasome in that its activity enhances the degradation of known unstable proteins like cyclins by upregulating proteasome activity directly (Raven and Koromilas, 2008).
Response: As explained in the first question, we might explore the specific mechanism of PERK kinase in the future studies.
- There are countless grammatical and vocabulary errors in the entire manuscript. For example, in lines 11-15, seven grammatical errors were found. The manuscript should be edited by someone with language training or a native English speaker prior to submission.
Response: We have made substantial changes on the manuscript. All grammatical and linguistic mistakes have been carefully checked and corrected in revised manuscript.
- 6. The Methods are poorly described and contain insufficient information on a procedure. Figure 4 utilizes sucrose gradient resolution of polysomes. The text says a 15-50% gradient was used, but the Methods say a 10-50% gradient was used. It says very little else. It is important to know what size gradient/SW rotor was used, how long it was centrifuged and at what g-force. The conditions might indicate that the author’s over-sedimented their gradients, and have pushed most polysomes over 7-mers to a pellet at the bottom of the tube. This may explain why all the mRNAs tested are low in fraction 14 and spike high in fraction 15.
Response:The sucrose gradients were performed in 10-50% sucrose. The rotor used was a SW41 and samples were spun at 35k rpm for 2h. We have included this in the text and modified the mistake in the text to reflect this. Since this is a sucrose density gradient, polysomes will stop migrating through the gradient when they reach the appropriate concentration of sucrose. Given the time and rpms/g-force used and that the observed peaks (80S, early polysomes) appear at normal places relative to other publications (reineke et al., 2018, zhu et al., 2019), we believe that the peak in later fractions is not an issue with respect to the conclusions we are making. Even if there were ribosomes pelleting, we still have sufficient resolution to see shifts in the mRNA.
- Treatment of cells prior to the polysome analysis is also problematic. The authors incubate cells 10 min in cycloheximide prior to lysis. This pushes even poorly translating mRNAs into dense, arrested polysomes and gives anomalously “heavy” polysomes, which is why their 1AT mRNA appears in 6-7-mer polysomes even when eIF2 is phosphorylated after MG treatment. It may also explain why puromycin, which should dissociate all polysomes when fed to cells, has just a modest effect on polysome size. The authors should also explain why both MG and puromycin cause the same reduction of 1AT polysome size (Fig. 4C) but MG lowers 1AT protein so much more than puromycin does (Fig 4A).
Response: Cycloheximide works by freezing elongating ribosomes. Thus, the ribosomes generally stall in the vicinity of where they were located on the mRNA at the start of the cycloheximide treatment. Any subsequent ribosomes that initiating translation thereafter are blocked by the stalled ribosomes. It is important to note that there are multiple mechanisms contributing to the location of the mRNAs here, and we have not completely teased out which mechanism is responsible for 1AT mRNA regulation. As the reviewer may know, many mRNAs are still translated under physiological eIF2a phosphorylation levels while others are repressed. For example, those with upstream open reading frames, like ATF4. mRNAs containing uORFs have been suggested to account for 50% of the mRNAs in the cell (Calvo et al., 2009, Iacono et al., 2005, Matsui et al., 2007). In acute conditions of eIF2a phosphorylation, where it is robustly phosphorylated, the majority of mRNAs are repressed (Reineke et al. 2018). During puromycin treatment, the degree of the shift is dependent on the kinetics of drug treatment. And each mRNA will respond differently especially in the linear phase of translation repression due to puromycin treatment. It is important to compare controls here. Bactin and 1AT both shift with puromycin as compared to their own controls. However, only 1AT shifts with MG132 as compared to its control suggesting it is more sensitive than Bactin, which does not shift with MG132 consistent with the polysome profiles. This is consistent with our assessment of MG132 stress as a physiological inducer of eIF2a phosphorylation. We also disagree that puromycin does not robustly inhibit protein synthesis (Figure 4B). We have calculated the polysome/monosome ratio in our samples, now shown in Fig4B, and this clearly indicates a strong shift with puromycin. The lower the number, the more repression.
- Some Methods are completely absent. For instance, the method used to compare 1AT mRNA levels by qRT-PCR in Fig 2A are completely absent. What mRNA was used as a standard to normalize? Did the authors use a 2EdeltadeltaCt formula, as is conventional?
Response: We added mRNA extraction and qPCR method in the methodology section in the revised manuscript.
References:
- Harding, H. P., Novoa, I., Zhang, Y., Zeng, H., Wek, R., Schapira, M. and Ron, D. (2000). Regulated translation initiation controls stress-induced gene expression in mammalian cells. Mol Cell 6, 1099-108.
- Walter, P. and Ron, D. (2011). The unfolded protein response: from stress pathway to homeostatic regulation. Science 334, 1081-1086.
- Raven, J. F. and Koromilas, A. E. (2008). PERK and PKR: old kinases learn new tricks. Cell Cycle 7, 1146-1150.
- Richter, J. D. and Smith, L. D. (1981). Differential capacity for translation and lack of competition between mRNAs that segregate to free and membrane-bound polysomes. Cell 27, 183-191.
Jiang HY, Wek RC (2005) Phosphorylation of the alpha-subunit of the eukaryotic initiation factor-2 (eIF2alpha) reduces protein synthesis and enhances apoptosis in response to proteasome inhibition. The Journal of biological chemistry 280: 14189-14202
Yerlikaya A, Kimball SR, Stanley BA (2008) Phosphorylation of eIF2alpha in response to 26S proteasome inhibition is mediated by the haem-regulated inhibitor (HRI) kinase. The Biochemical journal 412: 579-588

Reviewer 2 Report
In the manuscript by Rao et al., authors report a new role of an RNA binding protein Ras GAP SH3 binding protein (G3BP1) on the effects of proteasome inhibitors directed toward mRNA translation of alpha one antitrypsin (alpha1AT) in human hepatocytes. In addition, proteasome inhibition decreased protein synthesis in both human hepatocytes and macrophages. In particular, proteasome inhibitors are shown to increase the phosphorylation state of the translation initiation factor 2alpha. Conclusions are supported by elegantly executed experiments. Here are my suggestions:
- Some of the text in the introduction can be streamlined. Here is an example where streamlining can be done: “With a wide range of substrates, the proteasome has been implicated in regulation of important physiological processes such as the cell cycle and apoptosis. The proteasome has been extensively studied in the cancer field because of its regulatory role in cell cycle and apoptosis [15,16].”
- Authors should assess alpha1AT mRNA levels to exclude mRNA decay as an underlying reason to decreased protein levels.
- What is the half life-time of alpha1AT? alpha1AT protein seems to degrade very fast (within 4 hours), particularly given that proteasome is inhibited or downregulated. Authors should discuss this.
- Figure 1C,D; 7B: Authors should better explain what is mature medium alpha1AT presented in these panels.
- Figure 3A-C: authors should include better controls for S35 labeled proteins (e.g. selective proteins with unchanged synthesis along with no change in total protein synthesis).
- Figure 7B; b-actin western blot is unusual.
- Authors should include scatter plots with error bars in all figures to allow readers to see data distribution.
- Authors may want to cite (at minimum) following relevant articles: Alard et al., 2019 manuscript entitled “Differential Regulation of the Three Eukaryotic mRNA Translation Initiation Factor (eIF) 4Gs by the Proteasome”; Djabarkova et al., 2014 manuscript entitled “Translation Regulation and Proteasome Mediated Degradation Cooperate to Keep Stem-Loop Binding Protein Low in G1-phase”; Baugh and Pilipenko manuscript entitled “20S Proteasome differentially alters translation of different mRNAs via the cleavage of wIF4E and eIF3”.
- Authors may want to discuss possibility that RNA binding proteins can alter phosphorylation of initiation factors.
Author Response
In the manuscript by Rao et al., authors report a new role of an RNA binding protein Ras GAP SH3 binding protein (G3BP1) on the effects of proteasome inhibitors directed toward mRNA translation of alpha one antitrypsin (alpha1AT) in human hepatocytes. In addition, proteasome inhibition decreased protein synthesis in both human hepatocytes and macrophages. In particular, proteasome inhibitors are shown to increase the phosphorylation state of the translation initiation factor 2alpha. Conclusions are supported by elegantly executed experiments. Here are my suggestions:
1.Some of the text in the introduction can be streamlined. Here is an example where streamlining can be done: “With a wide range of substrates, the proteasome has been implicated in regulation of important physiological processes such as the cell cycle and apoptosis. The proteasome has been extensively studied in the cancer field because of its regulatory role in cell cycle and apoptosis [15,16].”
Response: We agree with the referee. We have taken the suggestion and made the amendment accordingly in revised manuscript. “As proteasome plays an essential role in regulation of important physiological processes such as the cell cycle and apoptosis, proteasome inhibitors has been extensively studied in the cancer field [15,16]. ”.
2.Authors should assess alpha1AT mRNA levels to exclude mRNA decay as an underlying reason to decreased protein levels.
Response: Agree with referee. We have dissected the alpha1AT mRNA mRNA degradation by northern blot Figure S2B. The result indicated 4hours’ treatment of MG132 does not change the alpha1AT mRNA level significantly.
3.What is the half life-time of alpha1AT? alpha1AT protein seems to degrade very fast (within 4 hours), particularly given that proteasome is inhibited or downregulated. Authors should discuss this.
Response: Alpha1AT is actually a very stable protein, a matured a1AT has a half-life of 4 to 5 days within blood circulation (Fregonese & Stolk, 2008). Our data Figure2B indicated over 4 hours of pulse-chasing period, less than 20% of cellular α1AT has been degraded also suggested α1AT is very stable intracellularly.
4.Figure 1C,D; 7B: Authors should better explain what is mature medium alpha1AT presented in these panels.
Response: We apologize for lack of clarity here. It should be mature α1AT found in cell culture medium. We have taken the suggestion and made new Figure 1 and7 with suggested amendments.
5.Figure 3A-C: authors should include better controls for S35 labeled proteins (e.g. selective proteins with unchanged synthesis along with no change in total protein synthesis).
Response: We apologize for lack of clarity here. Figure 3A and B showed α1AT has been down regulated by MG132 in translational level. Figure 3C is an experiment proves other proteasome inhibitors act in a similar way like MG132 in translational inhibition of a1AT.
MG132 treatment caused ER stress (Werner et al, 1996) and eif2a phosphorylation(Jiang & Wek, 2005) is well known, As eIF2α is a general translation initiation factor. It is very likely all protein synthesis could be affected by MG. However, we have Figure3D here to show that comparing with general protein synthesis inhibition, MG has an even more profound impact on a1AT translation.
6.Figure 7B; b-actin western blot is unusual.
Response: We made adjustment of the western blot figure. We are confident in our result.
7.Authors should include scatter plots with error bars in all figures to allow readers to see data distribution.
Response: We agree with the referee. We have made the amendment accordingly on Figure 2,3,5,7 and S3.
8.Authors may want to cite (at minimum) following relevant articles: Alard et al., 2019 manuscript entitled “Differential Regulation of the Three Eukaryotic mRNA Translation Initiation Factor (eIF) 4Gs by the Proteasome”; Djabarkova et al., 2014 manuscript entitled “Translation Regulation and Proteasome Mediated Degradation Cooperate to Keep Stem-Loop Binding Protein Low in G1-phase”; Baugh and Pilipenko manuscript entitled “20S Proteasome differentially alters translation of different mRNAs via the cleavage of wIF4E and eIF3”.
Response: We agree with the referee. We think the eif4E reference is relevant to our work, we have added the citation (Alard et al, 2019) in the revised manuscript.
9.Authors may want to discuss possibility that RNA binding proteins can alter phosphorylation of initiation factors.
Response: We agree with the referee. Actually, RNA binding proteins is able to change initiation factor like eiF2α. Over expression of RNA binding protein G3BP1 (also stress granule protein) is able to cause eiF2α phosphorylation by activate protein kinase (Reineke et al, 2012).
Alard A, Marboeuf C, Fabre B, Jean C, Martineau Y, Lopez F, Vende P, Poncet D, Schneider RJ, Bousquet C et al (2019) Differential Regulation of the Three Eukaryotic mRNA Translation Initiation Factor (eIF) 4Gs by the Proteasome. Front Genet 10: 254
Fregonese L, Stolk J (2008) Hereditary alpha-1-antitrypsin deficiency and its clinical consequences. Orphanet journal of rare diseases 3: 16
Jiang HY, Wek RC (2005) Phosphorylation of the alpha-subunit of the eukaryotic initiation factor-2 (eIF2alpha) reduces protein synthesis and enhances apoptosis in response to proteasome inhibition. The Journal of biological chemistry 280: 14189-14202
Reineke LC, Dougherty JD, Pierre P, Lloyd RE (2012) Large G3BP-induced granules trigger eIF2alpha phosphorylation. Molecular biology of the cell 23: 3499-3510
Werner ED, Brodsky JL, McCracken AA (1996) Proteasome-dependent endoplasmic reticulum-associated protein degradation: an unconventional route to a familiar fate. Proceedings of the National Academy of Sciences of the United States of America 93: 13797-13801

Reviewer 3 Report
The manuscript investigated the effect of proteasome inhibitor on the amount of the cellular and secretory α1-AT. The authors provided substantial experimental evidence to elucidate the mechanisms of the inhibitory effect of proteasome inhibitor on α1-AT. This manuscript includes many RNA related techniques. The manuscript, however, needs some polish and clarification on writing. Here are a few other things that may help to improve this manuscript:
Line 72 – 75. It is not clear what the author is trying to say.
Line 430 – 435. The pull down assay, was not very clear.
Line 452: typo “Ranse-free”
Table S1 “Full list of proteins binding with α1AT mRNA identified by MS” is not found in the manuscript, neither in the supplement file
Figure 1C and 1D, why there are α1-AT detection bands above the β-actin bands but also α1-AT detection bands underneath the β-actin bands? The two α1-AT looks very different?
Figure S1 A and S1 C, it seems the MG132 caused a high amount of GADD34. It would be nice if the authors could introduce what is GADD34 and why it is induced in high amount in MG132 treatment in the manuscript.
Figure 2C and Figure 3A seems contradictory. Figure 2C shows at 30 minutes, the secretory α1-AT is detected. But in Figure 3A shows in the medium, after 4 hours, the secretory α1-AT is not detected under MG132 treatment.
Figure 3B: in the figure legend mentioned “medium”, but there is no “medium” apeared in the figure. It only shows “cell lysate”.
Figure 6C: missing labeling in the figure. Based on Figure 6C, it is hard to tell that MG132 treatment causes the increase of RNA binding protein α1-AT because even with the transfection of the empty vectors, there are a decrease of α1-AT
It would be nice that the authors could note in the figure legend which are the western blot results and which are the northern blot results.
Author Response
The manuscript investigated the effect of proteasome inhibitor on the amount of the cellular and secretory α1-AT. The authors provided substantial experimental evidence to elucidate the mechanisms of the inhibitory effect of proteasome inhibitor on α1-AT. This manuscript includes many RNA related techniques. The manuscript, however, needs some polish and clarification on writing. Here are a few other things that may help to improve this manuscript:
Line 72 – 75. It is not clear what the author is trying to say.
Response: We apologize for the lack of clarity here. We have checked the manuscript thoroughly and corrected all the grammatical or other linguistic mistakes in revised manuscript.
.
Line 430 – 435. The pull down assay, was not very clear.
Response: We apologize for the lack of clarity here. We have added the IP procedure “The cells lysate with antibody were transferred into a 1.5 mL tube and rotated for overnight 4°C, and then added with 20 μl of Protein A/G Magnetic Beads (Thermo Fisher) and rotated for another 2 hours. The tube was then placed into a magnetic stand to collect the beads. The beads were washed 3 times using 400 μl lysis buffer and then resuspended into 30 μl elution buffer. The elution with target protein was adjusted to neutral pH with neutralization buffer and then applied on SDS-PAGE for protein separation. Finally, the gel was exposed to phosphor-imager to detect the radiation intensity. in the “Materials and Methods” section.
Line 452: typo “Ranse-free”
Response: We apologize for the typo. It should be “RNase-free”. We have made the amendment in revised manuscript.
Table S1 “Full list of proteins binding with α1AT mRNA identified by MS” is not found in the manuscript, neither in the supplement file
Response: We apologize for the mistake. We have added the protein list table1 in supplement.
Figure 1C and 1D, why there are α1-AT detection bands above the β-actin bands but also α1-AT detection bands underneath the β-actin bands? The two α1-AT looks very different?
Response: Thanks for bringing up this point. α1-AT is a protein composed of 414 amino acids with a molecular weight of 44 KDa. Before secreted outside hepatocytic cells, the protein is modified in ER by covalently linked with oligosaccharides. Thus, in the cell lysate, the two bands of signal represent the non-glycosylated (also called immature form with around 44kDa) and the glycosylated protein (also called mature form at around 52kDa) respectively.
As only the glycosylated a1AT could be secreted outside cells, only one band represent mature form of A1AT could be detected in cell culture medium.
Figure S1 A and S1 C, it seems the MG132 caused a high amount of GADD34. It would be nice if the authors could introduce what is GADD34 and why it is induced in high amount in MG132 treatment in the manuscript.
Response: We agree with the Referee.GADD34 is an unstable protein with short half-life. It is only sensitive to proteasome mediated digestion (Brush & Shenolikar, 2008). It is also transcriptionally induced by stress. It thus a good marker protein to indicate the proteasome inhibitor’s effect. We have added more introduction and discussion of GADD34 in revised manuscript.
Figure 2C and Figure 3A seems contradictory. Figure 2C shows at 30 minutes, the secretory α1-AT is detected. But in Figure 3A shows in the medium, after 4 hours, the secretory α1-AT is not detected under MG132 treatment.
Response: The lower paned of Figure 3A indicated the secreted α1-AT in the medium. As MG132 caused strong inhibition of α1-AT’s intracellular synthesis, it indirectly affects the mature form of glycosylated α1-AT’s production and secretion. Thus, a severe decrease of mature form of a1AT was found in 4hours treatment sample’s medium.
Figure 3B: in the figure legend mentioned “medium”, but there is no “medium” apeared in the figure. It only shows “cell lysate”.
Response: We apologize for the mistake. It should be “Brefeldin A (2 µg/ml) was added to the medium. α1AT was immunoprecipitated from cell lysates and [35S] labeled α1AT was detected by autoradiography”. We have made the amendment in revised manuscript.
Figure 6C: missing labeling in the figure. Based on Figure 6C, it is hard to tell that MG132 treatment causes the increase of RNA binding protein α1-AT because even with the transfection of the empty vectors, there are a decrease of α1-AT
Response: We apologize for the mistake. We fixed the labeling problem in the new Figure 6.
Over expression of G3BP can cause stress granule and translation inhibition(Reineke et al, 2012), thus in lane 3 even without MG132, the expression level of α1-AT is also reduced compared with non-transfected cells.
It would be nice that the authors could note in the figure legend which are the western blot results and which are the northern blot results.
Response: I apologize for the lack of clarity. We have made the modification accordingly in new Figure S2B.
Brush MH, Shenolikar S (2008) Control of cellular GADD34 levels by the 26S proteasome. Molecular and cellular biology 28: 6989-7000
Reineke LC, Dougherty JD, Pierre P, Lloyd RE (2012) Large G3BP-induced granules trigger eIF2alpha phosphorylation. Molecular biology of the cell 23: 3499-3510

Round 2
Reviewer 1 Report
The revised manuscript by Rao et al is much improved, particularly in grammar and clarity. Better descriptions of experiments, filling in missing methods, and more stringent descriptions of the Results make it more readable as well.
The authors now also acknowledge the potential regulation of a1AT mRNA by PERK via proteasome inactivation. However, they are reluctant to test it directly with an experiment such as PERK siRNA or use of Walcott Rallison PERK(-/-) cells (1). This would greatly improve enthusiasm for the conclusions they draw.
More concerning is that, while their rebuttal letter acknowledges that other mRNAs they have assayed directly (see below) for secreted proteins (possibly all?) are likely to be identically translationally suppressed by proteasome inhibitors, they are reluctant to state it in their manuscript. This gives the false impression that the a1AT mRNA is regulated uniquely, which is unlikely. The rebuttal letter states, "In fact, we agree that other mRNAs may be affected. We have also found other secreted proteins like album and α-fetoprotein are down regulated by MG132 treatment (figure bellow [sic. Fig was not included]), as our project is focus on the α1AD treatment, we didn’t include that in our discussion." A statement (if not the author's data) such as the one quoted here should appear in the Results relating to Figures 3 and/or 4, and be similarly evaluated in the Discussion. The interpretation of such findings differs substantially from the interpretation currently given in the Discussion. Otherwise, the paper is much improved.
1. PERK is required at the ER-mitochondrial contact sites to convey apoptosis after ROS-based ER stress.
Cell Death Differ. 2012 Nov;19(11):1880-91. doi: 10.1038/cdd.2012.74. Epub 2012 Jun 15. PMID: 22705852.Author Response
Please check the attachment file.
